# Role of vitamins in the pathogenesis and treatment of restless leg syndrome: A systematic review and meta-analysis

**Xiao-Min Xu**[1,2]*, **Jiang-hai Ruan**[1,2], **Tao Tao**[1,2], **Shu-li Xiang**[1,2], **Ren-liang Meng**[1,2], **Xiu Chen**[1,2]*

1 Department of Neurology, The Affiliated Hospital of Southwest Medical University, Lu zhou, Sichuan, China, 2 Laboratory of Neurological Diseases and Brain Function, Luzhou, Sichuan, China

* 706665270@qq.com (XC); xuxiaomin018167@163.com (XMX)

## Abstract

We performed a meta-analysis to assess the role of vitamins in the possible pathogenesis and treatment of RLS (Restless Leg Syndrome). A systematic search of the PubMed, Cochrane, Embase, and Web of Science databases was conducted. Fifty-nine studies on the relationship between vitamins and RLS were included, as well as four randomized controlled trials (RCTs) on vitamin treatment of RLS. Compared with non-RLS patients, serum vitamin D levels were significantly lower both in primary (P = 0.009) and secondary (P = 0.003) RLS patients, and appeared to be positively correlated with disease severity. Serum folate levels were lower in pregnant RLS patients than in pregnant non-RLS patients (P = 0.007), but this phenomenon was not seen in non-pregnant RLS patients (P = 0.65). Vitamin B12 (P = 0.59) and B1 (P = 0.362) deficiencies were not found in RLS patients. Oral vitamin B6 significantly improved primary RLS (P < 0.0001), while vitamin D did not (P = 0.05). Oral vitamin C (P < 0.00001), E (P < 0.0001), and vitamin C + E (P < 0.00001) all significantly improved hemodialysis-associated RLS with equal efficacy. Vitamin C is equivalent to 0.18 mg of pramipexole for the treatment of RLS (P = 0.81). In this meta-analysis, low vitamin D levels were found in patients with RLS, low folate levels were associated with RLS only in pregnant women, and vitamin C/E/B6 may improved symptoms in patients with RLS. These results suggest that vitamin deficiency or insufficiency may be related to the pathogenesis of RLS.

## Introduction

Restless leg syndrome (RLS) is a chronic neurosensorimotor disorder characterized by an irresistible urge to move the limbs, usually accompanied or caused by uncomfortable and unpleasant sensations, which occurs or worsens at rest and nighttime, and partially or completely relieved by movement [1–3]. The overall prevalence of RLS is about 7%, with 2.7% of the population having clinically significant RLS [4]. Genetic factors, dopaminergic dysfunction and brain iron deficiency are thought to be involved in the pathogenesis of RLS [2,5]. Primary RLS is often associated with familial inheritance and is thought to be related to genes such as *Meis1* and *BTBD9*. Secondary RLS is commonly seen in patients with renal

**Data availability statement:** All relevant data are within the paper and its Supporting Information files.

**Funding:** The author(s) received no specific funding for this work.

**Competing interests:** NO authors have competing interests.

**Abbreviations:** RLS, Restless leg syndrome; RCTs, Randomized controlled trials; MeSH, Medical Subject Headings; NOS, Newcastle Ottawa scale; MDs, Mean differences; CIs, Confidence intervals; ORs, Odds ratios; IRLS, International Restless Legs Scale; PSQI, Petersburg Sleep Quality Index.

failure, iron deficiency anemia, pregnancy, neurodegenerative diseases, or specific drug use [6,7]. Symptoms mainly affect the legs, but in severe cases can extend to the upper limbs [7]. It severely affects patients' sleep, induces emotional disorders, and reduces quality of life [6–8].

Vitamins are a class of organic substances necessary to maintain normal physiological functions of the human body, often a component of coenzymes, and play an indispensable role in the process of human growth, metabolism and development. Various clinical studies have investigated the relationship between vitamins and RLS, and the results showed that there are related changes in vitamin levels in patients with RLS, and supplementation of related vitamins can alleviate clinical symptoms. However, no meta-analysis or systematic reviews have been published discussing the role of vitamins in RLS. Therefore, we conducted a meta-analysis to evaluate the role of vitamins in the possible pathogenesis and treatment of RLS.

## Methods

### Data sources and search strategy

A systematic search of Cochrane, PubMed, Embase, and Web of Science databases from inception to January 2024 was conducted to search for matching trials. Keywords include such as 'restless legs syndrome*' and 'vitamin*', as detailed in the Supplemental Table 1. The search is performed using the following search keywords: "AND" and "OR" Boolean operators individually or in combination with one another. To avoid missing relevant controlled trials, the reference list of all relevant literature and relevant meeting abstracts were manually searched to further identify any potential studies eligible for inclusion. Only studies published in English were included. Search for all studies were conducted from January 10 to April 28, 2024. This systematic review was not registered in any database.

### Study selection

Import all retrieved publications into the literature management software EndNote. First, duplicate articles found were excluded, then the remaining literature that did not meet the inclusion criteria was excluded by reading the title and abstract, and finally the full text of the remaining literature was read to decide whether to include in the final analysis.

### Selection criteria

In order to comprehensively analyze the relationship between vitamins and RLS in existing studies, we intend to include all studies on the relationship between vitamin levels or vitamin intake and patients with RLS and the effects of vitamins on RLS,which we define as correlational studies and treatment studies, respectively.

Inclusion criteria for correlational studies: (i) clinical case-control trials or cross-sectional study or report; (ii) patients diagnosed with primary or secondary RLS, with no specific age limit; (iii) reported the association between vitamins and RLS, such as changes in vitamin levels in patients with RLS, or the relationship between vitamin intake and the onset of RLS; (iv) the main indicators can be obtained.

Inclusion criteria for treatment studies: (i) randomized controlled trials (RCTs); (ii) patients diagnosed with primary or secondary RLS, with no specific age or ethnic origin limit; (iii) vitamins taken as a treatment for RLS; if patients were also receiving a combination supplement of other nutrients and there was a control group in the study receiving the same other nutrients alone, we included them in the study as well; (iv) the main indicators can be obtained.

Exclusion criteria: (i) animal studies or in vitro studies; or (ii) reviews, case reports, dissertations, and duplicate analyses; or (iii) outcome data were incomplete or unavailable.

## Data extraction

Two authors (XMX, JHR) independently screened the title and abstract of each paper to verify that the study meets the inclusion criteria, and then review and validate the full text of the potential paper. Four reviewers (XMX, JHR, TT, SLX) independently extracted data from eligible studies. The extracted data included first author's name, publication date and country, RLS type, vitamin type, study design, study population, risk of bias, outcome data and associated factors. For correlational study, when the serum vitamin value is reported as nmol/L (pmol/L), we convert it to ng/mL (pg/ml), divided by the factor of 2.494 for 25-OH-VitD, 2.27 for folate, and 0.739 for vitamin B12. If any data could not be directly extracted, we chose to search associated conference summaries or other studies citing the RCT in question. Any disagreements regarding study inclusion and data extraction were resolved via discussion or following arbitration by the third reviewer (XC) if necessary.

## Quality assessment

The methodological quality of trials was assessed using the Risk of Bias Assessment Tool from the Cochrane Handbook for Systematic Reviews of Interventions [9]. Assessment included seven domains of bias (random sequence generation, allocation concealment, blinding, incomplete outcome data, selective reporting and other bias) on three grades(low, high, and unclear risk of bias).

The Newcastle Ottawa scale (NOS) was used to estimate the quality of the included datasets for the case-control study and cross-sectional study [10]. This assessment included three domains: selection,comparability, and outcome. Datasets are categorized as high, moderate, and low quality if scores are $\geq 7$, 4-6 and $\leq 3$, respectively.

Each study was evaluated by two independent reviewers, any disagreements were resolved via discussion or following arbitration by the third reviewer (XC) if necessary.

## Statistical analysis

All statistical analyses were conducted using RevMan 5.3 software (Cochrane Management System). Mean differences (MDs) with 95% confidence intervals (CIs) for continuous outcomes and odds ratios (ORs) with 95% CIs for dichotomous data. Heterogeneity was evaluated with $I^2$ statistics [9]. If a certain heterogeneity for outcome data was observed ($I^2 > 50\%$ or $P < 0.1$), a random-effects model was chosen to calculate pooled estimates. Otherwise, a fixed effect model was chosen. Where possible, meta-analyses of the intention-to-treat population were performed. $P < 0.05$ was defined as significant for heterogeneity. Subgroup analysis were performed based on the RLS type and study type. Sensitivity analysis was performed for vitamin correlation analysis, and the overall effect size of vitamin and RLS was re-analyzed after excluding tests with NOS scores as unclear. Over ten studies were evaluated for publication bias using funnel plot regression method. And Egger's tests were carried out in Stata version 18.0 statistics to confirm whether there is a small-study effect.

## Results

### Features of the included literature

An initial database search identified 1,746 articles (Pubmed: 147, Embase: 675; Cochrane: 397; Web of Science: 527), and a further four articles were identified through a manual search. Among them, 359 duplicate articles were eliminated by EndNote, 1391 by reading titles and abstracts, and 114 by reading the full text. Finally, 61 studies met the inclusion criteria and were included. 57 articles focused on the relationship between vitamins and RLS [11–69], and

4 randomized controlled trials focused on vitamin treatment of RLS [70–73]. Fig 1 outlines the screening process. Table 1 and Table 2 shows the characteristics of the included studies.

## Relationship between vitamins and RLS

Fifty-seven papers focused on the relationship between vitamins and RLS. These vitamins included vitamin D (n = 32), folate (n = 30), vitamin B12 (n = 24), vitamin B1 (n = 1), vitamin E (n = 1) and vitamin C (n = 1). Twenty-seven studies focused on primary RLS (mainly) and 30 studies focused on secondary RLS. Among the studies on secondary RLS, there were 12 studies on pregnancy with RLS and 8 studies on dialysis-related RLS.

**Vitamin D.**  Thirty-two studies have investigated the relationship between restless leg syndrome and vitamin D [11–42], including four on dialysis RLS [13,26,30,34] and four on pregnancy-related RLS [18,19,35,38]. The pooled analyses showed that RLS patients had significantly lower levels of vitamin D than controls (MD = -3.43; 95% CI = -5.29 to -1.57, P = 0.0003; heterogeneity, $I^2$ = 92%; P < 0.00001), whether primary (MD = -5.41; 95% CI = -9.48 to -1.34, P = 0.009; heterogeneity, $I^2$ = 96%; P < 0.00001) or secondary RLS (MD = -1.98; 95% CI = -3.29 to -0.66, P = 0.003; heterogeneity, $I^2$ = 65%; P = 0.0007) (Fig 2a), or in controlled (MD = -5.00; 95% CI = -8.77 to -1.23, P = 0.009; heterogeneity, $I^2$ = 95%; P < 0.00001) or cross-sectional studies (MD = -2.01; 95% CI = -3.41 to -0.61, P = 0.005; heterogeneity, $I^2$ = 68%; P = 0.0004) (Supplement Fig 1a); vitamin D deficiency/insufficient was detected in patients with RLS at a significantly higher rate than in controls without RLS (OR = 2.58; 95% CI = 1.87 to 3.56, P < 0.00001; heterogeneity, $I^2$ = 62%; P = 0.002), whether primary (OR = 4.49; 95% CI = 2.12 to 9.52, P < 0.0001; heterogeneity, $I^2$ = 78%; P = 0.0001) or secondary RLS (OR = 1.99; 95% CI = 1.61 to 2.46, P < 0.00001; heterogeneity, $I^2$ = 0%; P = 0.91) (Fig 2b), or in controlled (OR = 4.90; 95% CI = 2.09 to 11.49, P = 0.003; heterogeneity, $I^2$ = 74%; P = 0.0007) or cross-sectional studies (OR = 2.04; 95% CI = 1.72 to 2.42, P < 0.00001; heterogeneity, $I^2$ = 0%; P = 0.90) (Supplement Fig 1b); and patients with severe RLS had lower vitamin D levels than those with mild-moderate RLS (MD = -3.63; 95% CI = -6.19 to -1.07, P = 0.005; heterogeneity, $I^2$ = 0%; P = 0.42) (Fig 2c); RLS patients with deficient vitamin D levels (<20 ng/mL) had significantly higher IRLSSG scores than those with normal vitamin D levels (MD = 5.95; 95% CI = 0.75 to 11.14, P = 0.02; heterogeneity, $I^2$ = 96%; P < 0.00001) (Fig 2d); participants with vitamin D deficiency had a higher proportion of RLS detected than participants with normal vitamin D levels (OR = 2.81; 95% CI = 1.87 to 4.24, P < 0.00001; heterogeneity, $I^2$ = 27%; P = 0.25) (Fig 2e).

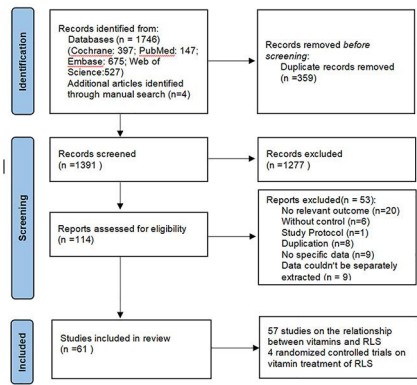

**Fig 1.  Flowchart of study selection.**

**Table 1. Clinical characteristics of articles on the correlation between vitamin and RLS.**

| Study | Location | Study type | RLS type | Sample | Technique | Vitamin type | Sample size(n) | Female% | Age (mean) | Test group vs. Control group | Quality |
|---|---|---|---|---|---|---|---|---|---|---|---|
| Botez 1977[43] | Canada | Cross-sectional | Prepregnancy (RLS) | Serum | L. CASEI | Folate | 11 vs 10 9 vs 12 | 100% | 29.19 | Oral multivitamin with 0.5g folic acid vs. without 0.5g folic acid; RLS vs. without RLS | unclear |
| O'Keeffe 1994[49] | Dublin | Case-control | Primary RLS (mainly) | Serum | NR | Vitamin B 12, Folate | 18 vs 16 | NR | 81 (median) | RLS vs. Controls | Moderate |
| Lee 2001[44] | USA | Cross-sectional | Prepregnancy (RLS) | Serum | NR | Folate | 7 vs 23 | 100% | 31.6 | Pregnancy with RLS vs. Pregnancy without RLS | Moderate |
| Manconi 2004[61] | Italy | Cross-sectional | Pregnancy (RLS) | NA | NA | Folate | 101 vs 445 | 100% | 31.7 | Pregnancy with new RLS vs. Pregnancy without RLS | Moderate |
| Högl 2005[63] | Italy | Cross-sectional | Primary RLS (mainly) | Blood | Standard techniques | Vitamin B 12, Folate | 74 vs 627 | 58.4% | 65.7 | RLS vs. Controls | High |
| Kemlink 2007[50] | Czech Republic | Cross-sectional | Familial + sporadic RLS | Blood | NR | Vitamin B 12, Folate | 43 vs 54 55 vs 92 | NR | NR | Familial RLS vs. Sporadic RLS | Moderate |
| Tunç 2007[64] | Turkey | Cross-sectional | Pregnancy (RLS) | Blood | NR | Vitamin B 12, Folate | 38 vs 108 | 100% | 24.81 | Pregnancy with RLS vs. Pregnancy without RLS | High |
| Manconi 2008[51] | Italy | Cross-sectional | MS (RLS) | Plasma | NR | Vitamin B 12, Folate | 34 vs 220; 35 vs 220 | NS | 18-65 | MS with RLS vs. MS without RLS | Moderate |
| Aksu 2009[52] | Turkey | Cross-sectional | Peritoneal dialysis (RLS) | Blood | NR | Folate | 13 vs 43 | NS | NS | Peritoneal dialysis with RLS vs. Peritoneal dialysis without RLS | Unclear |
| Kim 2010[65] | Korea | Cross-sectional | Primary RLS (mainly) | Blood | NR | Vitamin B 12, Folate | 59 vs 655 | 57.8% | 71.9 | RLS vs. Non-RLS | High |
| Suzuki 2011[62] | Japan | Cross-sectional | Migraine (RLS) | Blood | Standard clinical methods | Vitamin B1, B 12, Folate | 31 vs 179 | – | 14–72 | Migraine with RLS vs. Migraine without RLS | Moderate |
| Balaban 2012[14] | Turkey | Case-control | Primary RLS | Serum | ECLIA | 25(OH) D | 36 vs 38 | 74.3% | 38.85 | RLS vs. Healthy controls; | Moderate |
| Civi 2012[59] | Turkey | Cross-sectional | Primary RLS (mainly) | Blood | Standard techniques | Vitamin B 12, Folate | 53 vs 301 | 78.2% | 29.8 | RLS positive vs RLS negative | Moderate |
| Naini 2012[53] | Iran | Cross-sectional | Hemodialysis+Peritoneal dialysis (RLS) | Blood | NR | Vitamin B 12, Folate | 24 vs 66 | 41.1% | 54.2 | Hemodialysis with RLS vs. Hemodialysis without RLS | Moderate |
| Gade 2013[54] | Germany | Case-control | Hemodialysis (RLS) | Serum | Standard methods | Folate | 20 vs 26 | 69.6% | 67 | Hemodialysis with RLS vs. Hemodialysis without RLS | Moderate |
| Oran 2014[21] | Turkey | Cross-sectional | Primary RLS | Plasma | CMIA | 25(OH) D | 119 vs 36 | 90.3% | 49.32 | Vitamin D level < 20 ng/ml vs. Vitamin D level ≥ 20 ng/ml | High |
| Cakmak 2015[55] | Turkey | Cross-sectional | Primary RLS (mainly) | Serum | NR | Vitamin B 12, Folate | 169 vs 214 | 68.4% | ≥20 | RLS vs. Controls | High |
| Çakır 2015[20] | Turkey | Cross-sectional | Primary RLS | Blood | NR | Vitamin D | 57 vs 45 | 88.2% | 50.53 | Vitamin D level < 20 ng/ml (or with RLS) vs. Vitamin D level (or with RLS) ≥ 20 ng/ml | Moderate |

*(Continued)*

**Table 1.** (Continued)

| Study | Loca-tion | Study type | RLS type | Sample | Tech-nique | Vitamin type | Sample size(n) | Female% | Age (mean) | Test group vs. Control group | Quality |
|---|---|---|---|---|---|---|---|---|---|---|---|
| Cikrikcioglu 2016 [25] | Turkey | Case-control | Primary [+]/secondary(iron deficiency anemia) RLS | Serum | ELISA | 25(OH)D3 | 78 vs 78 | 100% | 45.7 | RLS vs. Controls | High |
| Demirci 2016[27] | Turkey | Cross-sectional | AS (RLS) | Blood | NR | Vitamin D, B12, Folate | 39 vs 69 | 29.6% | 39.5 | AS with RLS vs. AS without RLS | Moderate |
| Halac 2016[56] | Turkey | Case-control | Primary[+]/secondary RLS(iron deficiency anemia) | Blood | NR | Vitamin B 12, Folate | 37 vs 38 | 100% | 45.2 | RLS vs. Controls | Moderate |
| Helou 2016[67] | Lebanon | Cross-sectional | Haemodialysis (RLS) | Blood | NR | Vitamin C | 20 vs 27 vs 30 | NR | ≥18 | High vitamin C levels vs. Inter-mediate vitamin C levels vs. Low vitamin C levels | Unclear |
| Santos 2016[30] | Brazil | Cross-sectional | Hemodialysis (RLS) | Serum | Quimi-olu-mines-cence | 25(OH)D | 10 vs 9 | 68.4% | 48.0 | Hemodialysis with RLS vs. Hemo-dialysis without RLS | Moderate |
| Stefani 2016[41] | Austria | Case-control | Primary RLS (mainly) | Blood | NR | 25(OH)D | 57 vs 57 | NR | NR | RLS vs. Controls | Unclear |
| Varım 2016[57] | Turkey | Case-control | Primary RLS | Blood | NR | Vitamin B 12, Folate | 75 vs 56 | 73.3% | 46.67 | RLS vs. Controls | Moderate |
| Minár 2017[30] | Slovakia | Cross-sectional | MS (RLS) | Blood | NR | Vitamin D, B12, Folate | 52 vs.148 | 73.5% | 39.7 | MS with RLS vs. MS without RLS | High |
| Morker 2017[47] | USA | Cross-sectional | Prepregnancy (RLS) | Serum | NR | Folate | 20 vs 79 | 100% | 34.12 | Pregnancy with moderate or severe RLS vs. Pregnancy with mild or without RLS | Moderate |
| Atar 2017 [23] | Turkey | Cross-sectional | Primary RLS (mainly) | Serum | NR | 25(OH)D3 | 88 vs 122 | NR | 10-16 | Vitamin D deficient vs. Vitamin D insufficient and sufficient | unclear |
| Neves 2017[13] | Brazil | Case-control | Dialysis (RLS) | Serum | CLIA | 25(OH)D | 29 vs 72 16 vs 13 | 46.5% | 45.57 | Dialysis RLS vs. Dialysis control; Severe/very severe RLS vs. Mild/moderate RLS | High |
| Stefani 2017[16] | Austria | Case-control | Primary RLS (mainly) | Blood | NR | Vitamin D | 107 vs 107 | NR | NR | RLS vs. Healthy controls | Unclear |
| Becker 2018[58] | Ger-many | Cross-sectional | Inflammatory bowel disease (RLS) | Serum | NR | Folate | 5 vs 26 | 71.0% | ≥18 | With folate deficiency vs. Fithout folate deficiency | Moderate |
| Calviño 2018[31] | Spain | Cross-sectional | Renal trans-plant (RLS) | Serum | NR | 25(OH)D, B12,Fo-late | 19 vs 106 | 36.8% | 56.24 | Renal transplant with RLS vs. Renal transplant without RLS | High |
| Evans 2018[36] | India | Case-control | Primary RLS | Serum | NR | Vitamin D | 12 vs 13 37 vs 28 | NS | 3–12 years | RLS vs Controls; Mixed (GP/RLS) vs GP | Moderate |
| Huzmeli 2018[26] | Turkey | Cross-sectional | Hemodialysis (RLS) | Blood | NR | 25(OH)D | 33 vs 42 | 53.3% | 57.8 | Hemodialysis with RLS vs. Hemo-dialysis without RLS | Moderate |
| Işıkay 2018[28] | Turkey | Cross-sectional | Celiac disease (RLS) | Serum | NR | 25(OH)D, B12, Folate | 8 vs 218 | 46.9% | 13.24 | Celiac patients with RLS vs. Celiac patients without RLS | Moderate |

*(Continued)*

**Table 1.** (Continued)

| Study | Loca-tion | Study type | RLS type | Sample | Tech-nique | Vitamin type | Sample size(n) | Female% | Age (mean) | Test group vs. Control group | Quality |
|---|---|---|---|---|---|---|---|---|---|---|---|
| Wali 2018[12] | Saudi Arabia | Case-control | Primary + secondary RLS | Serum | NR | 25(OH)D, B12, Folate | 78 vs 123 59 vs 19 50 vs 28 | 51.7% | 44.38 | RLS vs. Healthy controls; RLS with vitamin D deficient vs. RLS with vitamin D sufficient; primary RLS vs. secondary RLS | High |
| Atalar 2019[22] | Turkey | Cross-sectional | Primary RLS | Serum | ELISA | 25(OH)D | 89 vs 63 | 46.7% | 46.13 | RLS with 25 (OH) D < 20 ng/mL vs. RLS with 25 (OH) D ≥ 20ng/mL | Moderate |
| Aynacı 2019[46] | Turkey | Cross-sectional | Prepregnancy (RLS) | NA | NA | Folate | 258 vs 66 | 100% | 29.18 | Folate intake ≥ 400 mcg/ day vs. Folate intake < 400 mcg/ day | Moderate |
| Bener 2019[29] | Turkey | Cross-sectional | T2DM (RLS) | Blood | NR | Vitamin D | 199 vs 672 | 64.4% | 50.33 | T2DM with RLS vs. T2DM without RLS | Moderate |
| SÜnter 2019[42] | Turkey | Cross-sectional | MS (RLS) | Blood | NR | Vitamin D, B12 | 30 vs 63 | 72% | 34.6 | MS with RLS vs. MS without RLS | Moderate |
| Tutuncu 2020[24] | Turkey | Cross-sectional | primary RLS | Serum | ECLIA | Vitamin D | 21 vs 11 | 43.8% | 45.06 | RLS with vitamin D deficiency vs. RLS with normal vitamin D level | Moderate |
| Almeneess-ier 2020[38] | Saudi Arabia | Cross-sectional | Pregnancy (RLS) | Serum | NR | 25(OH)D, Folate | 519 vs 223 | 100% | 29.2 | Pregnancy with RLS vs. Pregnancy without RLS | Moderate |
| Almeneess-ier 2020a [37] | Saudi Arabia | Cross-sectional | Primary+secondary(anemia/DM/CKD) RLS | Serum | NR | 25(OH)D | 271 vs 865 | 100% | 26.7 | RLS vs. Controls | Moderate |
| Çam 2020[66] | Turkey | Case-control | Primary RLS | Serum | NR | Vitamin B12 | 100 vs 106 | 85% | 41.7 | RLS vs. Controls | Moderate |
| Jiménez 2020[15] | Spain | Case-control | Primary RLS | Serum | ELISA | 25(OH)D | 111 vs 167 | 61.9% | 58.45 | RLS vs. Healthy controls | Moderate |
| Sağlam 2020[18] | Turkey | Cross-sectional | Pregnancy (RLS) | Serum | NR | 25(OH)D | 98 vs 47 57 vs 13 | 100% | 27.36 | Vitamin D level < 20 ng/ml(or with RLS) vs. Vitamin D level(or with RLS) ≥ 20ng/ml | Moderate |
| Andréas-son2021[60] | Sweden | Case-control | PD (RLS) | Serum | NR | Folate | 21 vs 21 | 28.6% | 69.3 | PD with RLS vs. PD without RLS | Moderate |
| Liu 2021[11] | China | Case-control | Primary RLS | Serum | Magnetic particle CLIA | 25(OH)D | 57 vs 57 21 vs 36 46 vs 11 | 68.4% | 57.62 | RLS vs. Healthy controls;(Extremely) severe RLS vs. Mild-moderate RLS; RLS with vitamin D insufficient vs. RLS with vitamin D normal | Moderate |
| Sarıcam 2021[39] | Turkey | Case-control | Primary RLS (with/without migraine) | Serum | NR | Vitamin D, B12 | 109 vs 105 | 71.5% | 43.7 | RLS (with/without migrain) vs. Controls | Moderate |
| Sun 2021[17] | China | Case-control | Primary RLS (including with migraine) | Serum | ECLIA | 25(OH)D | 49 vs 277 | 71.5% | 40.9 | RLS vs. Non-RLS control (including with migraine) | Moderate |
| Yalcinkaya 2021[40] | Turkey | Cross-sectional | MS (RLS) | Serum | NR | 25(OH)D | 11 vs 39 | 58% | 17.3 | MS with RLS vs. MS without RLS | Moderate |
| Geng 2022 [45] | china | Case-control | Primary RLS | Plasma | CMIA | Vitamin B 12, Folate | 80 vs 80 | 51.9% | 50.42 | Primary RLS vs. Healthy control | High |
| Alnaaim 2023[35] | Saudi Arabia | Cross-sectional | Pregnancy (RLS) | Blood | NA | Vitamin D | 122 vs 337 | 100% | ≥18 | Pregnancy with RLS vs. Pregnancy without RLS | Moderate |
| Marano 2023[33] | Italy | Cross-sectional | PD (RLS) | Serum | CLIA | 25(OH)D | 18 vs 32 | 34% | 69.5(median) | PD with RLS vs. PD without RLS | Moderate |

*(Continued)*

**Table 1.** (Continued)

| Study | Loca-tion | Study type | RLS type | Sample | Tech-nique | Vitamin type | Sample size(n) | Female% | Age (mean) | Test group vs. Control group | Quality |
|-------|-----------|------------|----------|--------|------------|--------------|----------------|---------|------------|------------------------------|---------|
| Miyazaki 2023[18] | Japan | Cross-sectional | Pregnancy (RLS) | Serum | LC-MS/MS,-CLEIA | 25(OH)D, Folate | 35 vs 168 | 100% | 32 | Pregnancy with RLS vs. Pregnancy without RLS | High |
| Das 2023[34] | Indian | Cross-sectional | CKD (RLS) | Serum | NR | Vitamin D | 11 vs 89 | 46% | 51.6 | CKD with RLS vs. CKD without RLS | Moderate |
| Turan 2023[48] | Turkey | Cross-sectional | Pregnancy (RLS) | Serum | NR | Vitamin B 12, Folate | 146 vs 354 | 100% | 27.8 | Pregnancy with RLS vs. Pregnancy without RLS | Moderate |

Abbreviations: RLS, restless leg syndrome; NR, not reported; NA, not applicable; MS, multiple sclerosis; ECLIA, electrochemi luminescence immunoassay; 25(OH)D, 25-hydroxy vitamin D; CMIA, chemiluminescence microparticle immunoassay; AS, ankylosing spondylitis; CLIA, chemiluminescent immunoassay; GP, growing pains; ELISA, enzyme linked immunosorbent assay; T2DM, Type 2 diabetes mellitus; DM, Diabetes mellitus; CKD, Chronic kidney disease; PD, Parkinson's disease; LC-MS/MS, liquid chromatography–tandem mass spectrometry.

**Table 2. Clinical characteristics of the included trails on vitamin therapy for RLS.**

| Study | Loca-tion | Study design | RLS type | Intervention | Dura-tion | Treatment group | Inter-vention vs place-bo(n) | Female % | Mean age | Out-come | Adverse effects (vitamins) |
|-------|-----------|--------------|----------|--------------|-----------|-----------------|------------------------------|----------|----------|----------|----------------------------|
| Jadidi 2023 [69] | Iran | Randomized, single-blind placebo-controlled | Pri-mary RLS | Vitamin B6 (40 mg, pill, daily) | 8 weeks | VitaminB6 + Pramipex-ole vs. Magnesium oxide (250mg) + Pramipexole vs.Placebo+pramipexole | 25 vs 25 vs 25 | 68% | 40.07 | IRLS score; PSQI score | NR |
| Rafie 2016 [70] | Iran | Randomized, double blind, placebo-controlled | Hemo-dialysis | Vitamin C (250 mg, tablet, daily) | 8 weeks | Vitamin C vs. Placebo vs. Pramipexole (0.18 mg) | 15 vs 15 vs 14 | 54.5% | 56.07 | IRLS score | No |
| Sagheb 2012 [71] | Iran | Randomized, double-blind, placebo-controlled | Hemo-dialysis | Vitamin C (200 mg,tablet,daily);Vi-tamin E(400mg,cap-sule,daily);Vitamin C+E | 8 weeks | Vitamins C + E vs. Vitamin C + placebo vs. Vitamin E + placebo vs. Double placebo | 15 vs 15 vs 15 vs 15 | 58.3% | 52.7 | IRLS score | Nausea, dyspepsia |
| Wali 2019 [68] | Saudi Ara-bia | Randomized, double-blind, placebo-controlled | Pri-mary RLS | Vitamin D (50,000 IU, caplets,weekly) | 12 weeks | Vitamin D vs. Placebo | 17 vs 18 | 31.4% | 42.55 | IRLS score | Abdominal pain,wors-ening of RLS symptoms |

Abbreviations: RLS, restless leg syndrome; IRLS, International Restless Legs Scale; PSQI, Petersburg Sleep Quality Index.

For dialysis-related RLS, there was significant difference in vitamin D levels between dialysis patients with RLS and those without RLS (MD = -4.88; 95% CI = -9.13 to -0.64, P = 0.02; heterogeneity, $I^2$ = 73%; P = 0.01) (Fig 2f); no significant difference in vitamin D levels between patients with severe/very severe RLS and mild/moderate RLS (MD = -6.40; 95% CI = -13.55 to 0.75, P = 0.08); no statistic difference in the detection rate of vitamin D deficiency (< 30 ng/mL) between dialysis patients with RLS and those without RLS (OR = 2.29; 95% CI = 0.95 to 5.55, P = 0.07). For pregnant women, the serum 25(OH)D level in the RLS group was lower than that in the non-RLS group (MD = -2.60; 95% CI = -4.26 to -0.94, P = 0.002); pregnant women with vitamin D deficiency had a higher proportion of RLS detected than those with normal vitamin D levels (OR = 1.96; 95% CI = 1.45 to 2.65, P < 0.0001); and RLS patients with vitamin D deficiency had significantly higher IRLSSG scores than those with normal vitamin D levels (MD = 6.40; 95% CI = 2.40 to 10.40, P = 0.002). However, there were

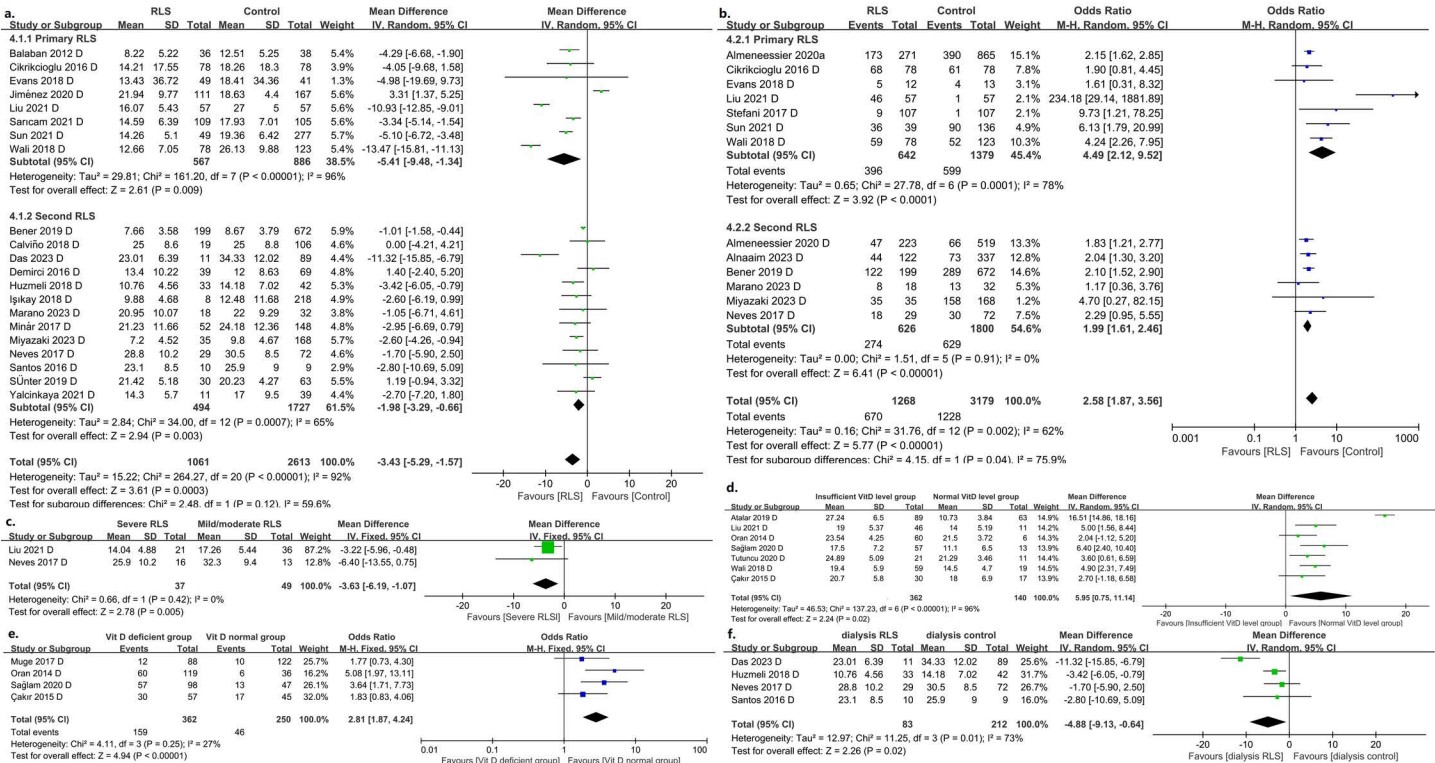

**Fig 2. a.** RLS patients had significantly lower levels of serum vitamin D than controls; **b.** The rate of serum vitamin D deficiency/insufficient in RLS patients was significantly higher than that in controls; **c.** Patients with severe RLS had lower vitamin D levels than those with mild-moderate RLS; **d.** RLS patients with deficient vitamin D levels had significantly higher IRLSSG scores than those with normal vitamin D levels; **e.** Participants with vitamin D deficiency had a higher proportion of RLS detected than participants with normal vitamin D levels; **f.** Dialysis patients with RLS had significantly lower levels of serum vitamin D than those without RLS.

also results showing no difference in vitamin D deficiency rates between pregnant women with and without RLS (OR = 4.70; 95% CI = 0.27 to 82.15, P = 0.29). However, except for the results of changes in vitamin D levels in dialysis patients from four studies, the other results above are from the conclusions of only one study.

**Folate.** Thirty articles examined the relationship between folate and RLS in patients [12,18,25,27,28,31,32,38,43–64], and nine articles focused on pregnancy-related RLS [18,38,43,44,46,47,49,62,64]. Of these articles, twenty-five looked at folate levels in patients with RLS (6 articles focused on pregnancy-related RLS) [12,18,25,27,28,31,32,38,44,45,47–54,56–58,60–62,64], four looked at the incidence of low folate in patients with RLS [18,55–57], and two looked at the incidence of RLS in patients taking folate [43,46]. The overall pooled analysis of the 25 studies showed that folate levels were significantly lower in pregnancy RLS women than in healthy pregnancy controls (MD = -5.30; 95% CI = -9.11 to -1.48, P = 0.007; heterogeneity,$I^2$ = 94%; P < 0.00001), while folate levels in non-pregnant RLS patients were not significantly different from controls (MD = 0.07; 95% CI = -0.24 to 0.38, P = 0.65; heterogeneity, $I^2$ = 4%; P = 0.41) (Fig 3a). Subgroup analysis based on the type of study showed that folate levels were lower in RLS patients in the cross-sectional trial than in the control group (MD = -2.15; 95% CI = -3.75 to -0.55, P = 0.009; heterogeneity, $I^2$ = 94%; P < 0.00001), but not in the case-control trial (MD = 0.15; 95% CI = -0.56 to 0.86, P = 0.009; heterogeneity, $I^2$ = 16%; P = 0.31) (Supplement Fig 1c). The rate of folate deficiency in pregnant (OR = 0.16; 95%

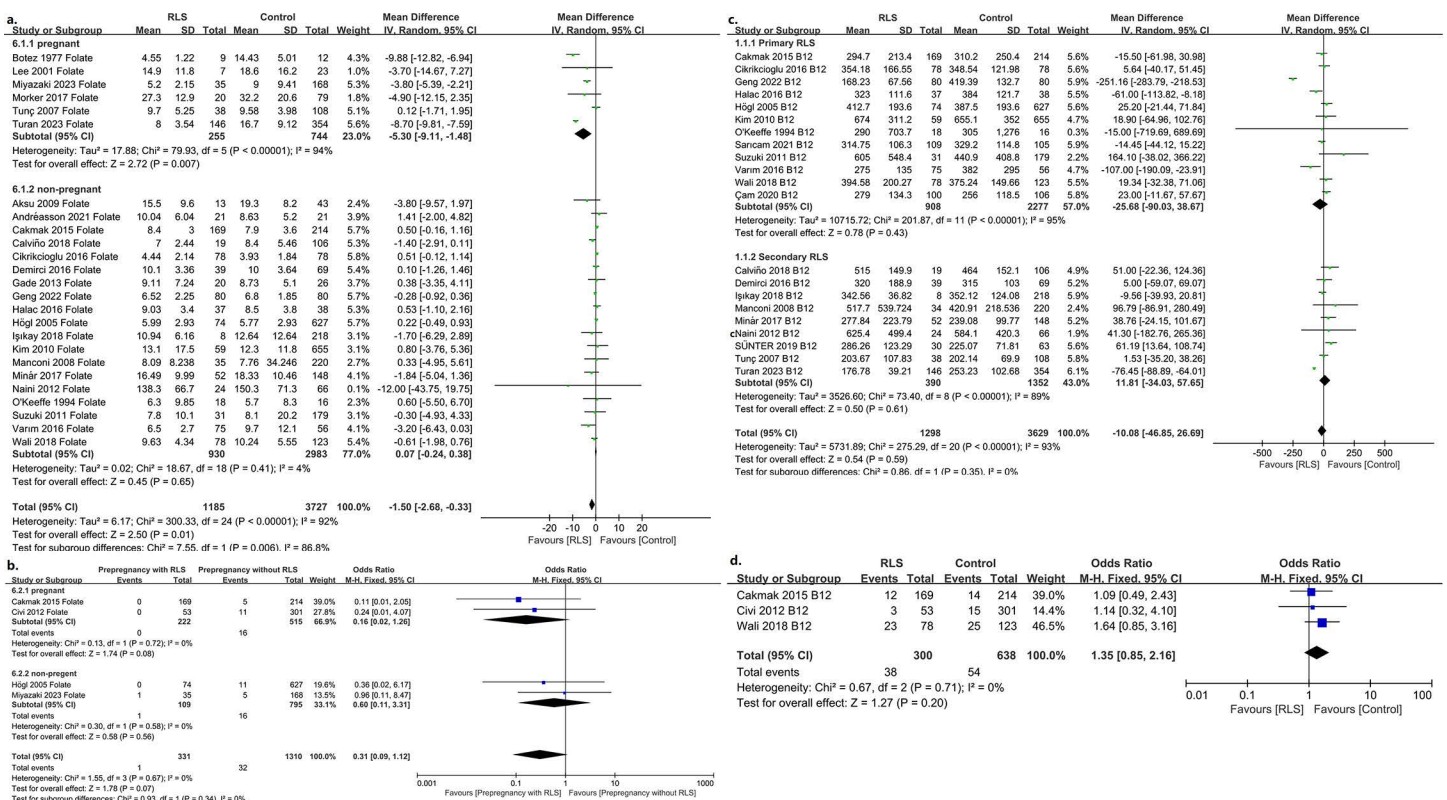

**Fig 3. a. Serum folate levels were significantly lower in pregnancy woman with RLS than those without RLS, while folate levels in non-pregnant RLS patients were not significantly different from controls** ; b. The rate of folate deficiency in pregnant or non-pregnant RLS patients was not different from that in the non-RLS controls; c. No statistical difference in serum vitamin B12 levels between patients with primary/secondary RLS and controls; d. No difference in the incidence of vitamin B12 deficiency between RLS patients and controls.

CI = 0.02 to 1.26, P = 0.08; heterogeneity, $I^2$ = 0%; P = 0.72) or non-pregnant (OR = 0.60; 95% CI = 0.11 to 3.31, P = 0.56; heterogeneity, $I^2$ = 0%; P = 0.58) RLS patients was not different from that in the non-RLS controls (OR = 0.31; 95% CI = 0.09 to 1.12, P = 0.07; heterogeneity, $I^2$ = 0%; P = 0.67) (Fig 3b). And pregnant participants with no (OR = 0.03; 95% CI = 0.00 to 0.33, P = 0.005) or insufficient folic acid intake (<400 mcg/ day) (OR = 0.55; 95% CI = 0.31 to 0.98, P = 0.04) had a higher incidence of RLS than those with adequate folic acid intake (≥ 400 mcg/ day). No difference in folate levels was found between sporadic and familial RLS (P = 0.48). Excluding articles that could not explicitly exclude women with a history of RLS prior to pregnancy, results from the three included studies showed no significant difference in the proportion (one study intake 0.5g/d folic acid, another one intake folic acid 0.8 mg/d + vitaminB12 4 ug/d) (OR = 0.99; 95% CI = 0.49 to 2.00, P = 0.98) and duration (MD = 0.00; 95% CI = -0.58 to 0.58, P = 1.00) of vitamin use before RLS occurred in pregnancy-related RLS patients compared with pregnant women without RLS.

**Vitamin B.** Twenty-one studies, including 1298 patients with RLS and 3629 controls, examined vitamin B12 levels and showed no statistical difference in vitamin B12 levels between patients with RLS and controls (MD = -10.08; 95% CI = -46.85 to 26.69, P = 0.59; heterogeneity, $I^2$ = 89%; P < 0.00001), whether primary (MD = -25.68; 95% CI = -90.03 to 38.67, P = 0.43; heterogeneity, $I^2$ = 95%; P < 0.00001) or secondary RLS (MD = 11.81; 95% CI = -34.03 to 57.65, P = 0.61; heterogeneity, $I^2$ = 89%; P < 0.00001) (Fig 3c), or in controlled

(MD = -54.26; 95% CI = -140.37 to 31.85, P = 0.22; heterogeneity, $I^2$ = 96%; P < 0.00001) or cross-sectional studies(MD = 14.52; 95% CI = -22.60 to 51.65, P = 0.44; heterogeneity, $I^2$ = 87%; P < 0.00001) (Supplement Fig 1d) [25,27,28,31,35,39,42,45,48,49,51,53,55–57,62–66,74]. In addition, there was no difference in the incidence of low vitamin B12 levels between the RLS and control groups from the three studies [12,55,59] (OR = 1.35; 95% CI = 0.85 to 2.16, P = 0.20; heterogeneity, $I^2$ = 0%; P = 0.71) (Fig 3d). One study involving vitamin B1 showed no difference in serum vitamin B1 levels between RLS patients and controls (P = 0.362) [62].

**Vitamin C/E.**  A conference paper described the relationship between vitamin C and the prevalence of RLS in dialysis patients [67]. The study included 77 adults who underwent hemodialysis for more than 6 months, 21 of whom had symptoms of RLS. It found that the incidence of RLS in dialysis patients was inversely proportional to vitamin C levels, but not related to vitamin E. Among them, the prevalence of RLS was zero in patients with high vitamin C level, 36.3% in patients with moderate vitamin C level, and 91.6% in patients with low vitamin C level (p = 0.041). This article was unable to obtain the definition of high or low vitamin C levels and the data related to vitamin E analysis.

## Sensitivity analysis

After excluding trials with NOS score as unclear (Stefani 2017 D, Botez 1977 Folate, Aksu 2009 Folate), Pooled estimates of effect sizes confirmed that the prevalence of vitamin D deficiency/deficiency was higher in patients with primary RLS than in controls without RLS (OR = 4.20; 95% CI = 1.91 to9.22, P = 0.0004; heterogeneity, $I^2$ = 81%; P < 0.0001) (Supplement Fig 2a), and that there was no difference in folate levels between non-pregnant RLS patients and controls (MD = 0.10; 95% CI = -0.20 to 0.39, P = 0.52; heterogeneity, $I^2$ = 0%; P = 0.46). The difference, however, was that folate levels in pregnant RLS women were lower than those in healthy pregnancy controls, but did not reach statistical significance (MD = -4.23; 95% CI = -8.57 to 0.11, P = 0.06; heterogeneity, $I^2$ = 94%; P < 0.00001) (Supplement Fig 2b).

## Vitamin treatment for RLS

Four papers on vitamin therapy for RLS have been published so far [70–73]. These vitamins include vitamin D, vitamin B6, vitamin C, vitamin E, and vitamin C + vitamin E. Two studies were conducted in patients with primary RLS and two studies in patients with hemodialysis-associated RLS.

**Vitamin D.**  Although several articles have been published examining the relationship between vitamin D and RLS, only one RCT has examined the effect of vitamin D on RLS [70]. In this trial, 35 individuals with primary mild to moderate RLS were randomized to receive either vitamin D or placebo, but only 22 participants completed the 12-week study. Compared with placebo, vitamin D did not reduce patients' RLS severity score (MD = 5.71; 95% CI = 0.12 to 11.30, P = 0.05) (Supplement Fig 3a) and remained true after data analysis was limited to RLS patients with vitamin D deficiency (MD = 7.99; 95% CI = -0.48 to 16.46, P = 0.06) (Supplement Fig 3b). The article also examined changes in vitamin D levels during treatment, and found that patients' restless leg symptoms did not decrease with significant increases in vitamin D levels. The study included a relatively small number of participants and did not provide information on whether patients were taking drugs such as pramipexole to treat RLS, as differences in these drugs between the two groups could have affected the trial results.

**Vitamin B6.**  A study published in 2023 explored the effect of vitamin B6 in alleviating the symptoms of restless legs syndrome [71]. The study showed that vitamin B6 significantly reduced IRLS scores and PSQI scores in RLS patients compared to placebo (IRLS scores: MD = -7.48; 95% CI = -11.19 to -3.77, P < 0.0001; PSQI scores: MD = -5.28; 95% CI = -6.91 to -3.65, P < 0.00001) (Supplement Fig 3c-d), comparable to magnesium oxide (IRLS scores: MD

= 0.60; 95% CI = -3.11 to 4.31, P = 0.75; PSQI scores: MD = 1.00; 95% CI = -0.51 to 2.51, P = 0.20) (Supplement Fig 3e-f).

**Vitamin C.** Two studies have explored the role of vitamin C in the treatment of RLS [72,73]. The pooled analyses showed that daily of vitamin C could significantly reduce IRLS scores in hemodialysis associated RLS patients compared to placebo (MD = -7.60; 95% CI = -9.70 to -5.50, P < 0.00001) (Supplement Fig 3g), comparable to pramipexole (0.18 mg) (MD = 0.59; 95% CI = -4.18 to 5.36, P = 0.81) (Supplement Fig 3h).

**Vitamin E.** Only one trial has examined the effect of vitamin E on RLS [73]. The trial showed that a daily vitamin E tablet significantly reduce symptoms in patients with hemodialysis associated RLS compared to a placebo (MD = -7.00; 95% CI = -10.39 to -3.61, P < 0.0001) (Supplement Fig 3i), and was equivalent to vitamin C (MD = -0.10; 95% CI = -3.62 to 3.42, P = 0.96) (Supplement Fig 3j).

**Vitamin C + E.** In the study conducted by Sagheb et.al [73], hemodialysis patients with RLS were randomly allocated to receive vitamin C and placebo, vitamin E and placebo, vitamin C + E, and double placebo for eight weeks. Compared with double placebo, vitamin C + E significantly reduced IRLS scores in patients (MD = -6.90; 95% CI = -9.23 to -4.57, P < 0.00001) (Supplement Fig 3k), but was not better than vitamin E + placebo(MD = -0.20; 95% CI = -4.25 to 3.85, P = 0.92) (Supplement Fig 3l) or vitamin C + placebo(MD = -0.30; 95% CI = -3.51 to 2.91, P = 0.85) (Supplement Fig 3m).

**Acceptability.** There was no statistical difference in discontinuation rates for any reason (p = 0.26) or for adverse events alone (p = 0.46) between the vitamin and control groups. Common adverse effects in the vitamin groups included nausea [73], dyspepsia [73], abdominal pain [70].

## Quality assessment and publication bias

The majority of the studies were ranked as moderate quality based on the NOS (**Supplemental Table 2**), and most treatment trials were at low risk of bias according to the Cochrane risk bias methods (supplement Fig 4a.4b). We evaluated studies on vitamin D levels (Egger's test P = 0.225), vitamin D deficiency/insufficient rate (Egger's test P = 0.078), folate levels (Egger's test P = 0.230), and vitamin B12 levels (Egger's test P = 0.063) for publication bias and found that the inverted funnel plots of the outcome data in these studies were basically symmetric(-supplement Fig 5a-d), and Egger's test also confirms this.

## Discussion

This meta-analysis provided an overview of the role of vitamins in RLS. Current research on vitamins related to RLS includes vitamins C, D, E, B1, B6, B12, and folate. Compared with non-RLS patients, serum vitamin D levels were significantly lower in both primary (P = 0.009) and secondary (P = 0.003) RLS patients, and appeared to be positively correlated with disease severity, with significantly higher rate of vitamin D deficiency/insufficient in RLS patients (P < 0.00001). Serum folate levels were lower in pregnant RLS patients than in pregnant non-RLS patients (P = 0.007), but this phenomenon was not seen in non-pregnant RLS patients (P = 0.65). Vitamin B12 (P = 0.59) and B1 (P = 0.362) deficiencies were not found in RLS patients. The incidence of RLS was inversely proportional to serum vitamin C and vitamin D levels and folate intake, but not to vitamin E levels. Oral vitamin B6 significantly improved primary RLS, while vitamin D did not. Oral vitamin C, E, and vitamin C + E all significantly improved hemodialysis-associated RLS with equal efficacy. Vitamin C is equivalent to 0.18 mg of pramipexole for the treatment of RLS. But it should be noted that to date, only single studies assessing the effect of vitamin supplementation on RLS have been conducted, making it difficult to draw definitive conclusions about their effectiveness.

The dysfunction of dopamine system is a key factor in the pathogenesis of RLS, which may increase the output of sympathetic neurons by impairing the descending modulation of the spinal circuit, thus altering the afferent input activity from muscle fibers, resulting in limb discomfort in patients [14]. Typical symptoms of RLS occur at night, which may be related to dopamine's circadian rhythm, which decreases at night and increases in the morning [68]. At the same time, serum iron levels have also been reported to drop by 50% during the night [44]. Iron is an essential element for the synthesis of dopamine, and reduced iron content in the thalamus, putamen, pallidum, and substantia nigra have been reported in RLS patients using magnetic resonance imaging phase analysis technology [21]. Therefore, the prevailing hypothesis is that RLS episodes may be the result of a variety of causes that impair iron homeostasis in the brain and thus interfere with dopamine synthesis [44,69].

Vitamin D is a fat-soluble vitamin and its action is mediated by the vitamin D receptor (VDR) [12,75]. Expression of vitamin D3 receptor protein has been demonstrated in midbrain dopamine neurons and their striatal target neurons [12]. Our statistical results showed that low serum vitamin D levels were significantly associated with RLS, both primary and secondary, suggesting a likely causal relationship between vitamin D and the occurrence of RLS. It has previously been reported that a decrease in dopamine levels in neonatal rats deficient in vitamin D, which subsequently affected the animals' behavior as adults, leading to increased locomotion [76]. Vitamin D has been found in vitro studies to increase the expression of tyrosine hydroxylase, a rate-limiting enzyme in dopamine synthesis, increasing the number of dopaminergic neurons [70,77], and increasing dopamine levels in the substantia nigra [70]. Previous studies have found that patients with iron deficiency anemia have low serum vitamin D levels [78], and patients with low ferritin concentration have low serum vitamin D concentration [22]. These associations may suggest that the interaction between low vitamin D and iron deficiency induces dysfunction of the dopaminergic system, thus inducing RLS. As can be seen, a decrease in vitamin D and the presence of iron deficiency anemia are often observed in women during pregnancy. The link between low vitamin D levels and RLS is well established, but the 12-week RCT of vitamin D for RLS did not show a positive result [70]. The possibility cannot be ruled out that the reasons are insufficient sample size included, the relatively short duration of treatment, and the dose and duration of vitamin D supplementation did not reach a level affecting the concentration of vitamin D in the brain. The role of vitamin D supplementation in the treatment of patients with RLS needs to be further clarified.

In patients with chronic renal insufficiency, antioxidant substances such as vitamin C and vitamin E decrease, while pro-oxidant activity increases, leading to increased oxidative stress. Hemodialysis may induce repeated oxidative stress responses mainly through membrane bioincompatibility and endotoxin challenge, aggravating the imbalance between pro-oxidation and antioxidant capacity [74]. Oxidative stress is thought to affect iron deposition, dopamine production, and thus promote RLS symptoms [4]. The results of the included literature found that the occurrence of RLS in dialysis patients may be related to the decrease of the antioxidant vitamin C level, but no association with vitamin E was found. Decreases in plasma vitamin C levels have been reported to be associated with short sleep [79], and it cannot be ruled out that further decreases in vitamin C levels are due to sleep loss associated with RLS. The role of vitamin C and vitamin E in reducing RLS symptoms may be due to their antioxidant properties, but the current findings do not suggest that combined supplementation of vitamins C and E is superior to taking them alone, although vitamin C may enhance the antioxidant function of vitamin E by regenerating a-tocopherol [73]. Previous in vitro studies have suggested that vitamin C and E can induce the production of tyrosine hydroxylase in neural cell lines and increase dopamine synthesis, and vitamin C can increase the absorption and utilization of iron, which may be other mechanisms by which these supplements are used

to treat uremic RLS [73]. Anyway, the current results suggest that vitamin C and E as antioxidants are safe and effective for the treatment of dialysis-associated RLS in the short term.

The average prevalence of RLS in pregnant women is 21%, about 2-3 times that of non-pregnant women [48,80,81]. In our study, thirteen articles have explored the relationship between vitamins and pregnancy-related restless leg syndrome, nine of which involved folate and four involved vitamin D. The results showed that the folate level of pregnancy-related RLS women was significantly lower than that of healthy pregnancy control group, but it was not found in non-pregnancy patients. The incidence of RLS in pregnant women with no folic acid intake or insufficient folic acid intake was higher than that in pregnant women with adequate folic acid intake (≥ 400 mcg/ day). Tetrahydrobiopterin is involved in dopamine synthesis as a cofactor of enzyme tyrosine hydroxylase, while folate plays a vital role in the regeneration of tetrahydrobiopterin [81]. If folate is reduced, dopamine synthesis may be restricted, contributing to the development of RLS. During pregnancy, levels of folate, iron, and ferritin often decrease due to the dilution of blood components, but not all pregnant women develop RLS during pregnancy, so it appears that susceptibility to RLS may be influenced or even determined by genetic background [81]. Studies have shown that familial RLS is more common in pregnant women with RLS than those without RLS, as well as other secondary RLS [80]. Our study showed that folate levels were significantly lower in patients with pregnancy-associated RLS compared to healthy pregnancy controls, but were not found to reach statistical significance after sensitivity analysis. Further large sample studies may be needed to verify this, and whether further declines in folate levels in pregnant women may be related to familial RLS requires further research.

The B vitamins are water-soluble vitamins that play an important role in cell metabolism. The RLS related B vitamins in this article include vitamin B1, B6 and B12. The results showed that vitamin B1 (P = 0.362) or B12 (P = 0.59) levels in RLS patients were not statistically different from those in the control group, and there were no controlled trials of vitamin B1 and B12 treatment for RLS; vitamin B6 significantly reduced IRLS score (P < 0.00001) and PSQI score (P = 0.20) in patients with RLS, but there is no literature to detect its change in RLS patients. For the effects of vitamin B6 in RLS, there is evidence that in addition to reducing plasma homocysteine, which is toxic to dopaminergic neurons, vitamin B6 has an antioxidant effect and thus a neuroprotective effect. Vitamin B6, on the other hand, is a cofactor of dopa carboxylase, an enzyme required for the conversion of levodopa into dopamine, which plays a key role in the synthesis of dopamine [82]. Previous studies also confirmed that the level of dopamine in the striatum corpus striatum of vitamin B6 deficient rats was significantly reduced [83] and the release time of dopamine into synapses was prolonged [84]. Therefore, vitamin B6 supplementation may improve the symptoms of RLS by improving dopamine, reducing homocysteine and antioxidant effects. Given the limited literature included in the study, the role of B vitamins in RLS remains to be defined.

Our article has several limitations. First, in the analysis of the association between vitamins and RLS, the existing studies included case-control and cross-sectional studies, but no cohort studies were found. Therefore, this analysis can only provide a possible correlation between vitamins and restless leg syndrome, and cannot determine a causal relationship between risk factors and restless leg syndrome, which is the inevitable limitation of conducting a meta-analysis based on the current research status. Second, due to the diversity of samples and sampling methods, differences in age and sex of participants, diversity of specimen testing methods, and various potential confounding factors, we obtained results with a high degree of heterogeneity, although subgroup analyses and sensitivity analyses have been performed to look for or reduce heterogeneity. Third, there are generally few articles on vitamin treatment of RLS, each vitamin may involve only one or two studies, and the sample size of the included studies is small, and the follow-up time is relatively short. Based on the limited overall data

available, the effectiveness and long-term effects of vitamins for the treatment of RLS cannot be well evaluated and determined.

## Conclusion

In this meta-analysis, low vitamin D levels were consistently found in patients with RLS, while low folate levels were observed specifically in pregnant women with RLS. In addition, based on limited data, vitamins C, E, and B6 may improved symptoms in patients with RLS. These results suggest that vitamin deficiency or insufficiency, particularly in vitamin D and folate, may be related to the pathogenesis of RLS. Considering that vitamin deficiency is a preventable and treatable disease, prospective cohort studies are warranted to determine the causal relationship with the onset of RLS and to further elucidate the mechanism of vitamins in RLS. The current preliminary data on vitamin therapy for RLS are promising, but further robust randomized controlled trials may be needed to validate their efficacy for RLS before these approaches can be routinely used for RLS.

## Supporting Information

**Supplement Fig 1. Subgroup analysis based on the study type. a. Vitamin D levels.** b. Rate of vitamin D deficiency/insufficient. c. Folate levels. d. Vitamin B12.
(TIF)

**Supplement Fig 2. Sensitivity analyses after excluding trials with NOS score as unclear. a.** Rate of vitamin D deficiency/insufficient. b. Folate levels.
(TIF)

**Supplement Fig 3. a-b.** Vitamin D did not reduce patients' RLS severity score compared to placebo, regardless of vitamin D deficiency; c-f. Vitamin B6 significantly reduced RLS patients' IRLS scores and PSQI scores compared to placebo, and was comparable to magnesium oxide; g-h. Oral vitamin C could significantly reduce IRLS scores in hemodialysis associated RLS patients compared to placebo, and was comparable to pramipexole; i-j. Oral vitamin E could significantly reduce IRLS scores in hemodialysis associated RLS patients compared to placebo, and was equivalent to vitamin C; k-m. Oral vitamin C + E significantly reduced IRLS scores in hemodialysis associated patient compared with double placebo, but was not better than vitamin E + placebo or vitamin C + placebo.
(TIF)

**Supplement Fig 4a. 'Risk of bias' summary: review authors' judgements about each risk of bias item for each included trials.** b. 'Risk of bias' graph: review authors' judgements about each risk of bias item presented as percentages across all included trials, with the majority of treatment trials having a low risk of bias.
(TIF)

**Supplemental Table 1. Results of the systematic search strategy.**
(DOCX)

**Supplemental Table 2. Results of the Newcastle-Ottawa scale assessment.**
(DOCX)

**Supplemental Table 3. Details of the information and data extracted from each study.**
(XLSX)

**Supplemental Table 4. Numbering table of all 1391 studies identified in the literature search and reasons for inclusion and exclusion.**
(DOCX)

## Acknowledgement

Xiu Chen and Xiao-min Xu are co-corresponding authors.

## Author contributions

**Conceptualization:** Xiao-min Xu, Xiu Chen.

**Data curation:** Xiao-min Xu, Jiang-hai Ruan, Tao Tao, Shu-li Xiang, Ren-liang Meng.

**Formal analysis:** Xiao-min Xu, Jiang-hai Ruan, Tao Tao.

**Investigation:** Xiao-min Xu, Xiu Chen.

**Methodology:** Xiao-min Xu, Jiang-hai Ruan, Tao Tao, Shu-li Xiang, Ren-liang Meng.

**Project administration:** Xiao-min Xu, Xiu Chen.

**Software:** Xiao-min Xu, Jiang-hai Ruan, Tao Tao, Shu-li Xiang, Ren-liang Meng.

**Supervision:** Xiu Chen.

**Validation:** Xiao-min Xu, Jiang-hai Ruan, Xiu Chen.

**Writing – original draft:** Xiao-min Xu, Tao Tao.

**Writing – review & editing:** Jiang-hai Ruan, Xiu Chen.

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
