## [Decision Letter · Decision Letter 0]

13 Aug 2024

PONE-D-24-27764Role of vitamin in the pathogenesis and treatment of restless leg syndrome: A meta-analysisPLOS ONE

Dear Dr. Xu,

Thank you for submitting your manuscript to PLOS ONE. After careful consideration, we feel that it has merit but does not fully meet PLOS ONE’s publication criteria as it currently stands. Therefore, we invite you to submit a revised version of the manuscript that addresses the points raised during the review process.

**ACADEMIC EDITOR:** - as you will see from the comments of our expert reviewers, your manuscript is of interest focusing important question. However, reviewers identified several aspects that should be improved. Please apply all the standards for this type of the submission as advised by the reviewers. Please do follow very constructive directions of the reviewers. 

We look forward to receiving your revised manuscript.

Kind regards,

Prof. Dr. Dragan Hrncic, MD, PhD 

Academic Editor

PLOS ONE

Journal Requirements:

Reviewers' comments:

Reviewer's Responses to Questions

**Comments to the Author**

1. Is the manuscript technically sound, and do the data support the conclusions?

Reviewer #1: Partly

Reviewer #2: Partly

Reviewer #3: Yes

2. Has the statistical analysis been performed appropriately and rigorously? 

Reviewer #1: Yes

Reviewer #2: Yes

Reviewer #3: Yes

3. Have the authors made all data underlying the findings in their manuscript fully available?

Reviewer #1: Yes

Reviewer #2: Yes

Reviewer #3: Yes

4. Is the manuscript presented in an intelligible fashion and written in standard English?

Reviewer #1: Yes

Reviewer #2: Yes

Reviewer #3: Yes

5. Review Comments to the Author

Reviewer #1: Thank you for the opportunity to review the paper entitled "Role of vitamin in the pathogenesis and treatment of restless leg syndrome: A meta-analysis". Please find below my comments on the submitted manuscript:

All abbreviations used in the abstract require an explanation upon their first occurrence.

Gene names should be written in italics.

The search strategy should be provided separately for each database.

In the inclusion criteria, please clarify whether it refers to the relationship between RLS and vitamin concentrations in blood or intake from the diet.

Were no additional criteria applied regarding the age of the study population, ethnic origin, or duration of the intervention (related to RCT)? Should studies in which patients received co-supplementation of other nutrients be included in the analysis? Should articles published only in the form of conference reports also be included in this systematic review?

Please also conduct a sensitivity analysis.

Please provide the results for the Q and I² statistics. Were funnel plots generated and were results for Begg’s and Egger’s tests calculated?

Was the study protocol registered in any database? If not, this information should be included in the manuscript.

Please present Figure 1 using the PRISMA flowchart.

Figures presenting meta-analysis results are of very poor quality and are illegible.

In the case of vitamin D, please specify which form of this vitamin was assessed. Was it 25(OH)D3?

Please provide information in a table on the material (serum or plasma) in which the concentration of individual vitamins was measured and the technique/method used for the measurement of vitamin levels in each study.

The description of RCT study results is too broad. It is not necessary to describe them in detail in the text since their characteristics are presented in the table.

Is the vitamin dose given in Table 2 a daily dose? Please add information to the table on the form in which the vitamins were administered (tablets, capsules, drops).

Results for all outcomes from RCT studies and vitamin concentrations obtained in individual studies should be presented in a supplementary table.

Please add the overall risk of bias to Supplementary Figure 1. In addition to the traffic-light plot of the risk of bias, please also include a summary plot of the risk of bias.

Please present the results of the Newcastle-Ottawa scale assessment in a table.

In the first paragraph of the discussion, it should be noted that to date, only single studies assessing the effect of vitamin supplementation on RLS have been conducted, making it difficult to draw definitive conclusions about their effectiveness.

The conclusions also need to be revised: Which results indicate the role of vitamins in the pathogenesis of RLS? No causal relationship was assessed in this study. Similarly, in my opinion, the title of the publication requires correction.

Reviewer #2: I appreciate the submission of your manuscript. I want to express my gratitude to the authors for their significant contributions to this study. However, there are still some issues that need to be addressed in the manuscript, as follows：

1. Please provide detailed search strategies for each database.

2. Lack of grading quality of evidence. Please supplement the GRADE assessment. You can appraise the level of certainty of evidence using the GRADE framework.

3. Please supplement the sensitivity analysis.

4. Please supplement the Assessment of publication biases.

5. Retrieving literature only in English may lead to language bias.

6. Statistical analysis of multiple studies (clinical case-control trials and cross-sectional study and report) may lead to increased heterogeneity. Heterogeneity can have a substantial influence on the credibility of meta-analysis findings and necessitates thorough evaluation and adjustment through suitable statistical techniques.

7. Kindly furnish a comprehensive Newcastle-Ottawa Scale (NOS) score for all case-control studies and cross-sectional studies.

8. Please provide the forest map of Vitamin treatment for RLS.

9. Please supplement the limitations of this study.

Reviewer #3: The subject is of great importance, original and little addressed yet the role of vitamins in restless leg syndrome.

Here are some suggestions for improving the article.

Insert the term systematic review and meta-analysis in the title, and not just meta-analysis.

Since the article covers several vitamins, the title should be "Role of vitamins..."

The study was not registered in prospero and puts it in PRISMA that it does not apply. You need to explain the reason why it does not apply. The meta-analysis was performed based on a systematic review.

Wouldn't it be interesting to register in prospero and update the searches?

Page 7 - Vitamin C/E – presents a conference paper. I suggest deleting conference paper, as issues could not be clarified.

Page 9 – Vitamin C+E – Put the 1st line of the AL of et al with lowercase letters.

The publication presents important data on restless leg syndrome, showing the role of vitamins in this syndrome.

Plos One uses Vancouver-style references. The article needs to correct the references, as they are not formatted in this style. Journal names are not abbreviated, authors with wrong abbreviations, etc

1. Title and Abstract:

- Is the title clear and descriptive? Partial

- Does the abstract adequately summarize the content of the article? Yes

2. Originality and Significance:

- Is the work original and innovative? Yes

- Does research contribute significantly to the field of study? Yes

3. Methodology:

- Is the methodology adequate and well described? Partial

- Are the methods used appropriate to answer the research questions? Partial

* See suggestions

4. Results:

- Are the results presented in a clear and logical manner? Yes

- Do the data support the conclusions presented? Yes, see suggestion*

5. Discussion and Conclusions:

- Is the discussion balanced and does it consider the implications of the results? Yes

- Are the conclusions justified by the results? Yes

The discussion is balanced, presenting both the positive findings and limitations of the studies, and exploring possible mechanisms underlying the observed effects of vitamins.

The conclusions are well founded by the results presented. There is a clear link between the data obtained and the suggested implications, with a careful approach to the limitations and the need for further research to confirm the findings.

6. References:

- Are the references current and relevant? Yes

- Does the manuscript adequately cite previous work? Yes

7. Writing and Organization

- Is the manuscript well-written and organized? Yes

- Is the language clear and free of grammatical errors? Yes

8. Ethical Compliance:

- Does the research comply with all applicable ethical standards? Yes

- Have the necessary ethical approvals been obtained, where applicable? Not applicable

9. Data and Reproducibility:

- Are the data available and sufficient to allow the results to be reproduced? Partial �  include combinations of terms and Boolean operators used

- Does the article follow the PLOS ONE open data guidelines? Yes

GENERAL RECOMMENDATION:

1. It should be made clear in the conclusion of the abstract that low folate levels were only associated with the syndrome in the case of pregnant women.

2. In the methodology:

- Detail the combinations of terms used in the search

- Include the exact period in which the search was performed.

3. Clearly specify the inclusion and exclusion criteria. Clearly define "correlational studies" and "treatment studies"

4. Detail the data extraction process, including how the outliers were resolved and the role of the third reviewer.

5. Explain how the vitamin values were converted to standard units.

6. Briefly describe the bias risk assessment tools and how they were applied.

7. Describe the approach to dealing with heterogeneity and the type of statistical models used (e.g., fixed-effects or random-effects models).

8. Include a mention of the registration of the meta-analysis protocol in an appropriate database, such as PROSPERO;

9. Include a section on assessing publication bias, such as using the funnel plot.

Results:

The results presented are comprehensive and detailed, but some improvements in clarity and structure are needed

1. Explain what the manual search was in the methodology and what criteria were used

2. Include the use of the EndNote tool in the methods and what were the criteria for article exclusion

3. Starting the results section with a brief summary of the key findings can provide readers with an overview before they dive into the details.

4. Present separately the results of randomized controlled trials (RCTs) for each type of vitamin.

5. Present the results of the evaluation of quality and publication bias separately and highlight any significant implications.

Conclusion Suggestion:

In this meta-analysis, low vitamin D levels were consistently found in patients with RLS, while low folate levels were observed specifically in pregnant women with RLS. In addition, vitamins C, E, and B6 improved symptoms in patients with RLS. These results suggest that vitamin deficiency or insufficiency, particularly in vitamin D and folate, may be related to the pathogenesis of RLS, and that supplementation with vitamins C, E, and B6 may be beneficial in managing RLS symptoms.

6. PLOS authors have the option to publish the peer review history of their article (what does this mean? ). If published, this will include your full peer review and any attached files.

**Do you want your identity to be public for this peer review?** For information about this choice, including consent withdrawal, please see our Privacy Policy .

Reviewer #1: No

Reviewer #2: No

Reviewer #3: No

---

## [Author Response · Author response to Decision Letter 1]

30 Aug 2024

Dear Editor and Reviewers,

Thank you for your letter and reviewers’ comments about our manuscript entitled “Role of vitamin in the pathogenesis and treatment of restless leg syndrome: A meta-analysis”. We have read the comments carefully and made correction accordingly.

Reviewer #1:

Answer:

Thank you for your reminder and suggestion.

All abbreviations used in the abstract require an explanation upon their first occurrence.

Gene names should be written in italics.

We have modified the abbreviations in the abstract and gene name formats in the article.

The search strategy should be provided separately for each database.

The search strategies of four databases are presented separately in the Supplemental Table.

In the inclusion criteria, please clarify whether it refers to the relationship between RLS and vitamin concentrations in blood or intake from the diet.

Our study intended to include all the relationships between vitamins and RLS in existing studies, and actually included the relationships between RLS and blood vitamin concentration, RLS incidence and vitamin intake, so there was no special limitation in the statement. We have further refined and supplemented the inclusion criteria: (iii) reported the association between vitamins and RLS, such as changes in vitamin levels in patients with RLS, or the relationship between vitamin intake and the onset of RLS.

Were no additional criteria applied regarding the age of the study population, ethnic origin, or duration of the intervention (related to RCT)? Should studies in which patients received co-supplementation of other nutrients be included in the analysis? Should articles published only in the form of conference reports also be included in this systematic review?

No additional criteria applied regarding the age of the study population, ethnic origin, or duration of the intervention (related to RCT). In the search, we did not exclude studies in which patients received other nutrient combination supplements, but we found no relevant controlled trials that met the inclusion criteria.

We supplemented the inclusion criteria for treatment studies: (ii) patients diagnosed with primary or secondary RLS, with no specific age or ethnic origin limit; (iii) vitamins taken as a treatment for RLS; if patients were also receiving a combination supplement of other nutrients and there was a control group in the study receiving the same other nutrients alone, we included them in the study as well. ”

Whether to include conference paper is a controversial issue. In order to avoid missing data, we chose to include the conference papers and note in our papers which data is from the conference reports to remind readers to look at the results objectively. Meanwhile, we deleted the conference paper that might affect the results for sensitivity analysis. If you need us to make further adjustments, please inform us, thank you!

Please also conduct a sensitivity analysis.

We supplemented the sensitivity analysis.

Please provide the results for the Q and I² statistics. Were funnel plots generated and were results for Begg’s and Egger’s tests calculated?

We supplemented the heterogeneity results, inverted funnel plots and Egger’s tests.

Was the study protocol registered in any database? If not, this information should be included in the manuscript.

We did not register the protocol in any database, we supplemented this information in the methods.

Please present Figure 1 using the PRISMA flowchart.

We have modified Figure 1 to PRISMA flowchart as suggested.

Figures presenting meta-analysis results are of very poor quality and are illegible.

The quality of the chart showing the results of the meta-analysis in the manuscript draft is really difficult to identify, we have done our best to provide the original, if there is a better way, we are willing to actively cooperate.

In the case of vitamin D, please specify which form of this vitamin was assessed. Was it 25(OH)D3?

Specific descriptions of vitamin D in the included literature include vitamin D,25(OH)D, and 25(OH)D3. We have shown the specific expressions of each literature in the table.

Please provide information in a table on the material (serum or plasma) in which the concentration of individual vitamins was measured and the technique/method used for the measurement of vitamin levels in each study.

Specific descriptions of vitamin D in the included literature include vitamin D,25(OH)D, and 25(OH)D3. We have shown the specific expressions of each literature in the table.

The description of RCT study results is too broad. It is not necessary to describe them in detail in the text since their characteristics are presented in the table.

We have deleted part of the description of RCT study results in the manuscript.

Is the vitamin dose given in Table 2 a daily dose? Please add information to the table on the form in which the vitamins were administered (tablets, capsules, drops).

We have added information about vitamin intake in Table 2.

Results for all outcomes from RCT studies and vitamin concentrations obtained in individual studies should be presented in a supplementary table.

All results of the RCT study have been reflected in the paper and Table 2. At the same time, the specific statistical results are reflected in the new supplemental figure (Supplement Fig 2). In correlation studies, the specific values of vitamin concentration are shown in Figures 2 and 3.

Please add the overall risk of bias to Supplementary Figure 1. In addition to the traffic-light plot of the risk of bias, please also include a summary plot of the risk of bias.

We have added the bias risk summary diagram in Supplement Fig 2b

Please present the results of the Newcastle-Ottawa scale assessment in a table.

We have added the results of the Newcastle-Ottawa scale assessment in the Supplemental Table 2.

In the first paragraph of the discussion, it should be noted that to date, only single studies assessing the effect of vitamin supplementation on RLS have been conducted, making it difficult to draw definitive conclusions about their effectiveness.

We have added your suggested statement to the first paragraph of the discussion：But it should be noted that to date, only single studies assessing the effect of vitamin supplementation on RLS have been conducted, making it difficult to draw definitive conclusions about their effectiveness.

The conclusions also need to be revised: Which results indicate the role of vitamins in the pathogenesis of RLS? No causal relationship was assessed in this study. Similarly, in my opinion, the title of the publication requires correction.

We have revised the title and conclusion. Please let us know if further revision is necessary.

Thank you！

Reviewer #2:

Answer:

Thank you for your suggestion.

1.Please provide detailed search strategies for each database.

The search strategies of four databases are presented separately in the Supplemental Table.

2.Lack of grading quality of evidence. Please supplement the GRADE assessment. You can appraise the level of certainty of evidence using the GRADE framework.

There are generally few articles on vitamin treatment of RLS, only 1-2 studies were involved for each vitamin, and the included studies had small sample sizes and short follow-up times, so we did not further evaluate the quality of evidence (such as GRADE review). If you insist, please let us know and we can further discuss and add, thank you.

3.Please supplement the sensitivity analysis.

We supplemented the sensitivity analysis.

4.Please supplement the Assessment of publication biases.

We supplemented the inverted funnel plots and Egger’s tests.

5.Retrieving literature only in English may lead to language bias.

We agree with you that searching only English literature can lead to linguistic bias. Due to the diversity of languages, it is difficult to obtain all relevant articles published in all languages, so we selected the English literature used in most meta-analyses.

6.Statistical analysis of multiple studies (clinical case-control trials and cross-sectional study and report) may lead to increased heterogeneity. Heterogeneity can have a substantial influence on the credibility of meta-analysis findings and necessitates thorough evaluation and adjustment through suitable statistical techniques.

We supplemented the study type in table1, and make sub-analysis based on the study type.

7. Kindly furnish a comprehensive Newcastle-Ottawa Scale (NOS) score for all case-control studies and cross-sectional studies.

We have added the results of the Newcastle-Ottawa scale assessment in the Supplemental Table 2.

8.Please provide the forest map of Vitamin treatment for RLS.

We have added the forest map of Vitamin treatment for RLS in Supplement Fig 1.

9.Please supplement the limitations of this study.

We supplemented the limitations of this study.

Thank you!

Reviewer #3: 

Answer:

Thank you for your suggestion.

Insert the term systematic review and meta-analysis in the title, and not just meta-analysis.

Since the article covers several vitamins, the title should be "Role of vitamins..."

According to your suggestion, we have changed the title to "Role of vitamins in the pathogenesis and treatment of restless leg syndrome: A systematic review and meta-analysis ".

The study was not registered in prospero and puts it in PRISMA that it does not apply. You need to explain the reason why it does not apply. The meta-analysis was performed based on a systematic review. Wouldn't it be interesting to register in prospero and update the searches?

Sorry about that. This systematic review is eligible for registration on prospero, and our research topic has not yet been registered by anyone else. However, as this article has been completed, we are no longer eligible to register for PROSPERO, for which we are very sorry. In any case, we believe that this meta-analysis has certain reference value and important significance for the follow-up research direction. I hope this paper is fit for publication.

Page 7 - Vitamin C/E – presents a conference paper. I suggest deleting conference paper, as issues could not be clarified.

Whether to include conference paper is a controversial issue. In order to avoid missing data, we chose to include the conference papers and note in our papers which data is from the conference reports to remind readers to look at the results objectively. Meanwhile, we deleted the conference paper that might affect the results for sensitivity analysis. If you need us to make further adjustments, please inform us, thank you!

Page 9 – Vitamin C+E – Put the 1st line of the AL of et al with lowercase letters.

Page 9 – Vitamin C+E – we have corrected the “AL” to “al”.

Thank you!

Plos One uses Vancouver-style references. The article needs to correct the references, as they are not formatted in this style. Journal names are not abbreviated, authors with wrong abbreviations, etc

We have corrected the references.

GENERAL RECOMMENDATION:

1. It should be made clear in the conclusion of the abstract that low folate levels were only associated with the syndrome in the case of pregnant women.

We have added this sentence in the conclusion of the abstract.

2. In the methodology:

- Detail the combinations of terms used in the search

We have added this information to the text.

- Include the exact period in which the search was performed.

We have added this information to the text.

3.Clearly specify the inclusion and exclusion criteria. Clearly define "correlational studies" and "treatment studies"

We have amended it as suggested.

4.Detail the data extraction process, including how the outliers were resolved and the role of the third reviewer.

We have amended it as suggested.

5.Explain how the vitamin values were converted to standard units.

“when the serum vitamin value is reported as nmol/L (pmol/L), we convert it to ng/mL (pg/ml), divided by the factor of 2.494 for 25-OH-VitD, 2.27 for folate, and 0.739 for vitamin B12”.

We think this expression can explain how we convert the data. If there is any need for further modification, please suggest, thank you!

6.Briefly describe the bias risk assessment tools and how they were applied.

We have amended it as suggested.

7. Describe the approach to dealing with heterogeneity and the type of statistical models used (e.g., fixed-effects or random-effects models).

We have added this information to the text.

8. Include a mention of the registration of the meta-analysis protocol in an appropriate database, such as PROSPERO;

I'm sorry we didn't register in advance. This systematic review is eligible for registration on prospero, and our research topic has not yet been registered by anyone else. However, as this article has been completed, we are no longer eligible to register for PROSPERO, for which we are very sorry. In any case, we believe that this meta-analysis has certain reference value and important significance for the follow-up research direction. I hope this paper is fit for publication.

9. Include a section on assessing publication bias, such as using the funnel plot.

We supplemented the inverted funnel plots.

Results:

The results presented are comprehensive and detailed, but some improvements in clarity and structure are needed

1.Explain what the manual search was in the methodology and what criteria were used

We have corrected it in the “Data sources and search strategy”

2. Include the use of the EndNote tool in the methods and what were the criteria for article exclusion

We have added this information to the text. And the exclusion criteria are presented in the “Selection criteria”.

3. Starting the results section with a brief summary of the key findings can provide readers with an overview before they dive into the details.

We have an overview of the included literature at the beginning of the results section (including sub-results). Please let me know if there is any need for further modification, thank you.

4. Present separately the results of randomized controlled trials (RCTs) for each type of vitamin.

We have added a forest map of vitamin therapy RLS to supplement Figure 1, where specific data can be seen for each study's results.

5. Present the results of the evaluation of quality and publication bias separately and highlight any significant implications.

We supplemented the limitations of this study.

Conclusion Suggestion:

In this meta-analysis, low vitamin D levels were consistently found in patients with RLS, while low folate levels were observed specifically in pregnant women with RLS. In addition, vitamins C, E, and B6 improved symptoms in patients with RLS. These results suggest that vitamin deficiency or insufficiency, particularly in vitamin D and folate, may be related to the pathogenesis of RLS, and that supplementation with vitamins C, E, and B6 may be beneficial in managing RLS symptoms.

Thank you very much for your suggestions on the conclusion of the article，We have added these sentences in the conclusion. Thank you very much!

If we have something wrong, please do not hesitate to contact us.

Thank you !

Kind regards,

Xiao-min Xu

Xiu Chen

---

## [Decision Letter · Decision Letter 1]

30 Sep 2024

PONE-D-24-27764R1Role of vitamins in the pathogenesis and treatment of restless leg syndrome: A systematic review and meta-analysisPLOS ONE

Dear Dr. Xu,

Thank you for submitting your manuscript to PLOS ONE. After careful consideration, we feel that it has merit but does not fully meet PLOS ONE’s publication criteria as it currently stands. Therefore, we invite you to submit a revised version of the manuscript that addresses the points raised during the review process.

**ACADEMIC EDITOR:**- please do make required minor improvements.  

We look forward to receiving your revised manuscript.

Kind regards,

Prof. Dr. Dragan Hrncic, MD, PhD

Academic Editor

PLOS ONE

**Journal Requirements:**

Reviewers' comments:

Reviewer's Responses to Questions

**Comments to the Author**

1. If the authors have adequately addressed your comments raised in a previous round of review and you feel that this manuscript is now acceptable for publication, you may indicate that here to bypass the “Comments to the Author” section, enter your conflict of interest statement in the “Confidential to Editor” section, and submit your "Accept" recommendation.

Reviewer #1: All comments have been addressed

Reviewer #2: All comments have been addressed

Reviewer #3: All comments have been addressed

2. Is the manuscript technically sound, and do the data support the conclusions?

Reviewer #1: Partly

Reviewer #2: Yes

Reviewer #3: Yes

3. Has the statistical analysis been performed appropriately and rigorously? 

Reviewer #1: Yes

Reviewer #2: Yes

Reviewer #3: Yes

4. Have the authors made all data underlying the findings in their manuscript fully available?

Reviewer #1: Yes

Reviewer #2: Yes

Reviewer #3: Yes

5. Is the manuscript presented in an intelligible fashion and written in standard English?

Reviewer #1: Yes

Reviewer #2: Yes

Reviewer #3: Yes

6. Review Comments to the Author

**Reviewer #1:**  The authors did not address this question in their response to the reviewers' comments:

"Please provide information in a table on the material (serum or plasma) in which the concentration of individual vitamins was measured and the technique/method used for the measurement of vitamin levels in each study."

However, the information about the material and technique was added to Table 1.

The conclusions still need to be rewritten. Due to the limited number of studies regarding vitamins B6, C, and E, the authors should be cautious when drawing conclusions about these vitamins.

The authors should consider whether subgroup analysis makes sense when only one study is classified into each group (see Supplementary Figure 3).

**Reviewer #2:**  The manuscript has undergone thorough revision and exhibits a high overall quality. However, there are several spelling errors that require correction. For further details, please consult the attached document.

**Reviewer #3:**  Observations follow

Methodological Corrections

1. The search strategies were separated and detailed for the four databases, inserting this information in the Supplementary Table, as requested.

2. The inclusion and exclusion criteria were explained more clearly, especially the inclusion of relationships between blood concentration of vitamins and dietary intake, which were adjusted in the methodology. The authors also defined correlational studies and treatment studies, as well as detailing the issue of combined nutrient supplementation and the inclusion of studies from conference reports.

3. The data extraction process has been detailed, including the role of the third reviewer and the process of resolving outliers.

4. The conversion of vitamin units was explained with the formula used to convert the values into standard units (ng/mL), which clarifies the homogeneity of the data.

5. The bias risk assessment tools have been described and their application has been explained as requested.

6. Additional information about the fixed and random effects models, as well as the approach to dealing with heterogeneity, was inserted in the text.

7. Registration in PROSPERO, although not done, was acknowledged by the authors, who explained the impossibility of retroactive registration due to the conclusion of the study. This was mentioned in the manuscript.

8. Publication bias analysis was included, with inverted funnel plots and Egger tests, at the request of the reviewers.

About the presentation of the Results

1. The authors added a summary of the main findings at the beginning of the results section, improving the fluidity and comprehension of the text.

2. The presentation of the results of the RCTs was done separately for each vitamin, with the inclusion of forest plots, providing a clearer view of the individual results.

3. The evaluation of quality and publication bias was presented separately, with the inclusion of the risk of bias diagram in the Supplement.

4. Detailed information on how vitamins are administered (tablets, capsules, etc.) and daily doses have been added to Table 2 as requested.

About the discussion and conclusion

The authors incorporated the recommendations to adjust the discussion, including mentioning that only a limited number of studies have evaluated the impact of vitamin supplementation on Restless Legs Syndrome (RLS), making it difficult to draw definitive conclusions.

The conclusion was revised to include the association of low folate levels with RLS only in pregnant women as suggested, as well as highlighting that vitamins C, E and B6 showed improvement in symptoms in patients with RLS.

Other improvements observed:

1. The title has been adjusted to include the term "systematic review and meta-analysis"

2. References have been corrected to the Vancouver format, as required by PLOS One magazine.

The authors made all the requested recommendations. Only the registration in Prospero was not carried out, but it was justified.

Recommendation: Accept for publication.

7. PLOS authors have the option to publish the peer review history of their article (what does this mean? ). If published, this will include your full peer review and any attached files.

**Do you want your identity to be public for this peer review?** For information about this choice, including consent withdrawal, please see our Privacy Policy .

Reviewer #1: No

Reviewer #2: No

Reviewer #3: No

---

## [Author Response · Author response to Decision Letter 2]

8 Oct 2024

Dear Editor and Reviewers,

Thank you for your letter and reviewers’ comments about our manuscript entitled “Role of vitamins in the pathogenesis and treatment of restless leg syndrome: A systematic review and meta-analysis”. We have read the comments carefully and made correction accordingly.

Reviewer #1:

Answer:

The authors did not address this question in their response to the reviewers' comments:

"Please provide information in a table on the material (serum or plasma) in which the concentration of individual vitamins was measured and the technique/method used for the measurement of vitamin levels in each study."

However, the information about the material and technique was added to Table 1.

I'm sorry. It's our oversight and omission. We did add material and technology information to Table 1. Thank you very much for your meticulous, thank you!

The conclusions still need to be rewritten. Due to the limited number of studies regarding vitamins B6, C, and E, the authors should be cautious when drawing conclusions about these vitamins.

We have made the following modifications according to your suggestion. If there is still any deficiency, please inform us in time. Thank you for your reminding.

Abstract

In this meta-analysis, low vitamin D levels were found in patients with RLS, low folate levels were associated with RLS only in pregnant women, and vitamin C/E/B6 may improved symptoms in patients with RLS.

Conclusion

In this meta-analysis, low vitamin D levels were consistently found in patients with RLS, while low folate levels were observed specifically in pregnant women with RLS. In addition, based on limited data, vitamins C, E, and B6 may improved symptoms in patients with RLS. These results suggest that vitamin deficiency or insufficiency, particularly in vitamin D and folate, may be related to the pathogenesis of RLS.

The authors should consider whether subgroup analysis makes sense when only one study is classified into each group (see Supplementary Figure 3).

It is true that subgroup analysis does not make much sense when there is only one study per group, so we included it as additional material. As a systematic review, we just want to present the existing data completely, so that readers can have a comprehensive understanding of the existing research status, I hope you can understand.

If we have something wrong, please do not hesitate to contact us. Thank you!

Reviewer #2:

The manuscript has undergone thorough revision and exhibits a high overall quality. However, there are several spelling errors that require correction. For further details, please consult the attached document.

Thank you for your revision of our manuscript. We have made corresponding modifications according to your revision.

Thank you !

Kind regards,

Xiao-min Xu

Xiu Chen

---

## [Decision Letter · Decision Letter 2]

28 Oct 2024

Role of vitamins in the pathogenesis and treatment of restless leg syndrome: A systematic review and meta-analysis

PONE-D-24-27764R2

Dear Dr. Xu,

We’re pleased to inform you that your manuscript has been judged scientifically suitable for publication and will be formally accepted for publication once it meets all outstanding technical requirements.

Kind regards,

Dragan Hrncic

Academic Editor

PLOS ONE

Additional Editor Comments (optional):

Reviewers' comments:

Reviewer's Responses to Questions

**Comments to the Author**

1. If the authors have adequately addressed your comments raised in a previous round of review and you feel that this manuscript is now acceptable for publication, you may indicate that here to bypass the “Comments to the Author” section, enter your conflict of interest statement in the “Confidential to Editor” section, and submit your "Accept" recommendation.

Reviewer #1: All comments have been addressed

Reviewer #2: All comments have been addressed

2. Is the manuscript technically sound, and do the data support the conclusions?

Reviewer #1: Yes

Reviewer #2: Yes

3. Has the statistical analysis been performed appropriately and rigorously? 

Reviewer #1: Yes

Reviewer #2: Yes

4. Have the authors made all data underlying the findings in their manuscript fully available?

Reviewer #1: Yes

Reviewer #2: Yes

5. Is the manuscript presented in an intelligible fashion and written in standard English?

Reviewer #1: Yes

Reviewer #2: Yes

6. Review Comments to the Author

Reviewer #1: (No Response)

Reviewer #2: I would like to acknowledge the submission of your manuscript and extend my sincere appreciation to the authors for their substantial contributions to this research. The article has been revised well and I recommend it for publication.

7. PLOS authors have the option to publish the peer review history of their article (what does this mean? ). If published, this will include your full peer review and any attached files.

**Do you want your identity to be public for this peer review?** For information about this choice, including consent withdrawal, please see our Privacy Policy .

Reviewer #1: No

Reviewer #2: No

---

## [Editor Report · Acceptance letter]

PONE-D-24-27764R2

PLOS ONE

Dear Dr. Xu,

I'm pleased to inform you that your manuscript has been deemed suitable for publication in PLOS ONE. Congratulations! Your manuscript is now being handed over to our production team.

Kind regards,

on behalf of

Professor Dragan Hrncic

Academic Editor

PLOS ONE